# Biogas Production from Residues of Industrial Insect Protein Production from Black Soldier Fly Larvae *Hermetia illucens* (L.): An Evaluation of Different Insect Frass Samples

**Harald Wedwitschka [1],\*, Daniela Gallegos Ibanez [1] and Damián Reyes Jáquez [2]**

[1] Department Biochemical Conversion, DBFZ Deutsches Biomasseforschungszentrum Gemeinnützige GmbH, Torgauer Straße 116, D-04347 Leipzig, Germany
[2] Unidad de Posgrado, Investigación y Desarrollo Tecnológico, Tecnológico Nacional de México, Instituto Tecnológico de Durango, Victoria de Durango 34080, Mexico
\* Correspondence: harald.wedwitschka@dbfz.de; Tel.: +49-341-2434-562

**Abstract:** Insect biomass shows promise as an alternative animal feedstuff with a low climate effect. Industrial insect rearing generates residual materials, such as feed remains and insect excrements, so-called insect frass, which exhibits a high organic content. Commonly, these residues are utilized as soil amendment. Information on the suitability of these residues for biogas production is rather scarce. The energetic utilization of insect frass as feedstock for anaerobic digestion (AD) would allow for the simultaneous residue material reduction and bioenergy production. Additionally, synergies in heat management could arise using the exhaust heat of the biogas plant in the insect farming process. In laboratory-scale anaerobic digestion trials, the specific methane yield (SMY) of six different insect frass samples from black soldier fly (Hermetia) rearing were tested in batch biochemical methane potential (BMP) tests. Further, semi continuous anaerobic digestion trials on a lab scale using continuously stirred tank reactors (CSTRs) were carried out with Hermetia insect frass from a pilot plant operation in order to determine the digestibility and process stability of the AD process. The BMP results showed SMY values of the different insect frass samples ranging from $201 \pm 9$ to $287 \pm 37$ mL/gVS that are similar to those of other animal excrements, such as cow or pig manure already been used as feedstock in agricultural biogas plants. Results of the semi-continuous digestion of insect frass from the pilot plant operation showed a SMY value of $167 \pm 15$ mL/gVS, suggesting no process-inhibiting effect caused by the feed material. Although, the high nitrogen content must be taken into account for stable AD performance.

**Keywords:** insect frass; black soldier fly; *Hermetia illucens*; anaerobic digestion; BMP tests; CSTR digestion





## 1. Introduction

The continued growth in global meat production is leading to an increasing demand for high-quality protein feed. Due to the limited availability of natural resources, increasing climate change, and land-use competition between food-feed-fuel production, the importance of cost-effective and sustainably produced protein sources is growing [1].

The utilization of insects as feed animals, as food sources for human nutrition and for the production of technical products, such as silk, shellac, or bee wax, has a long tradition. Insects are the most diverse class of animals, with approximately one million described species. Due to their evolutionary development history, they are optimally adapted to a variety of habitats, environmental conditions, and feed materials. They are able to convert a wide spectrum of organic substrates and residues into high-quality raw materials.

Industrially produced insect meal represents an alternative feed protein source [2] and has been successfully tested as livestock, pet, and aquaculture feed [3–8]. Insect biomass shows a high protein content and a high-quality amino acid spectrum.

Nutritionally, insect protein is also suitable for human consumption and the climate impact of insect protein production turns out to be better compared to conventional animal protein production. Water consumption, land requirements, and required feed quantities and slaughter losses are generally lower in insect farming than in cattle and pig fattening and fish production. In Europe, however, the consumption of insects is hardly culturally anchored and consumer acceptance of insects as food is comparatively low. In contrast, there are hardly any reservations about the use of insects as animal feed [9]. Forecasts see the largest market potential for insects as feed in the aquaculture and pet food sectors, followed by livestock feed for poultry and swine [10].

In addition, insect products can be used as a bio-based alternative to conventional fossil raw materials in the production of a wide range of technical products, such as cosmetics, pharmaceuticals, surfactants, surface coatings, lubricants, and fuels. Insect farming represents a promising building block of a future bioeconomy, because against the background of limited resources, the multiple use and recycling of biomass in utilization cascades is increasingly gaining in importance. The carbon footprint of insect products is particularly advantageous when residual materials are used as insect feed and the process heat requirement is covered by exhaust heat or with the aid of renewable energies. Another advantage of insect protein production is that the water and land requirements and the amount of feed used for insect farming are relatively low, and residual materials can be returned to the nutrient cycle, for example, as biogas substrate or as agricultural fertilizer [11,12].

Insect framing of black soldier fly, respectively Hermetia in industrial scale comprises of the process steps fly rearing, larva fattening, and product processing [13]. The insect *Hermatia illucens* goes through the following development stages: egg, larvae, pupae, and fly, whereby insect protein is usually derived from the adult larvae. Fly breeding requires light and warmth and is necessary in order to provide sufficient young larvae for the fattening process. The fattening of Hermetia larvae is carried out without light, commonly in boxes and tubs of different sizes in a climate-controlled environment. In this production step, young larvae are added to a feed medium. Hermetia larvae consume organic matter, such as carbohydrates, proteins, and lipids, and increase in weight. Before entering the pupae state, larvae are separated mechanically from feed residues and insect excrement. The majority of the larvae are further processed into animal feed, while a smaller proportion is used for further fly rearing. Normally, all production steps take place in centralized farm concepts which ideally contribute from favorable heat energy and feed material supply. Decentralized concepts with outsourced larval fattening and centralized larva processing and product recovery represent an alternative approach.

The feed remains and insect excrements are residue materials of the Hermetia farming process and are usually utilized as organic fertilizers and soil additives [14]. According to European law, insect frass may only be used as agricultural fertilizer after sufficient sanitization. This requires heating to at least 70 °C for at least one hour. Possible hygienization measures that can meet the temperature requirements would be, for example, the heating, pelleting, or extrusion of the insect frass. All of these processes require a relatively high energy input but ensure that neither pathogens nor live larvae are released into the environment. Another technology that is used for waste biomass treatment and has a proven sanitization effect is the biogas process [15–18].

With regards to insect frass, anaerobic digestion is an interesting waste treatment option providing bio-methane as biofuel or an energy source for the production electricity and heat which could be reused in the insect rearing process and product processing. Additionally, residues of the digestion process still hold plant nutrients contained in the substrate material and can be utilized as organic fertilizer and soil amendment. The process combination of the biogas plant and insect farm enables various synergies. By integrating the insect farm into existing biogas plants, the exhaust heat utilization of the biogas plant could be optimized and digested residues from the biogas plant could also serve as an insects feed source [19].

Insect farming as the first stage of corresponding value chains and the utilization of residues from insect production in the biogas process could increase the efficiency of resource utilization. The large-scale production of Hermetia is a new technology. There are so far only a few companies worldwide with an insect production capacity on an industrial scale. Data on the methane potential of the residue materials are scarce and there are no data available on the long-term digestibility when this manuscript was written. One research aim of the study was the assessment of the biomethane potential of insect frass from Hermetia rearing on different feed sources. Insect frass samples were subjected to biochemical methane potential (BMP) tests in triplicate in laboratory batch scale in order to determine the specific methane yields and methane production kinetics of the sample materials. A further aim of the study was to evaluate the feedstock suitability of insect frass for AD processes in long term semi-continuous anaerobic digestion trials. Therefore, digestion experiments on a lab scale were carried out with actual insect frass from a large-scale pilot production of Hermetia in order to determine the feedstock digestibility and AD process stability. The AD characteristics presented in this study extend the data basis required for the suitability assessment of Hermetia insect frass as a raw material for biogas production. The BMP results determined can be used for an economic feasibility evaluation of the energetic utilization of insect frass in the AD process.

## 2. Materials and Methods

### 2.1. Acquisition and Characterization of the Insect Frass Samples

Five insect frass samples were obtained from previous Hermetia rearing trials in laboratory scale run between 2020 and 2021, where Hermetia larva were fed on five different substrates: corn silage (CS), brewers spend grain (BS), thin stillage from bioethanol production (ST), aquatic plants from Elodea genus (EL), and bran (BR). Larvae feeding trials were carried out in triplicate batch attempts in 550 mL plastic containers (CLIP & CLOSE Food storage container, EMSA, Germany) with a size of $16.3 \times 11.3 \times 5.8$ cm and a working volume of approximately 250 mL. The container caps were perforated to allow gas exchange. The substrate feed load was 240 mg VS/Larvae. Containers were stored at $30.0 \pm 0.25$ °C in temperature chambers (New Brunswick Innova 44). After 12 days, the feed remains and excrements (insect frass) were separated from the larvae and used in the present study without prior drying. In addition, another sample of insect frass was obtained from Hermetia rearing in pilot plant operation (IF_PP) (Hermetia Baruth GmbH, Baruth Mark, Germany). In 2020 and 2021, the annual production capacity of the pilot plant was approximately 300 t Hermetia larvae which were fed on a feed mixture mainly composed of cereal grain.

Insect frass samples were tested for their material properties regarding total solids (TS) and volatile solids (VS), nitrogen, protein, fat, and fibre composition and subjected to biochemical methane potential (BMP) tests. Additionally, long term semi-continuous digestion experiments were performed with the insect frass sample from a pilot plant operation (IF_PP). Wet samples were stored at 5 °C after sampling. Dry samples were stored in air tight plastic barrels at room temperature. Sample characteristics are depicted in Table 1.

**Table 1.** Insect frass sample material characteristics.

| Insect Frass | TS * | VS * | Ash * | Crude Protein | Raw Fat | Crude Fibre | Other Carbohydrates |
|---|---|---|---|---|---|---|---|
| (Feedstock) | [%FM] | [%TS] | [g/kgTS] | [g/kgTS] | [g/kgTS] | [g/kgTS] | [g/kgTS] |
| Stillage (ST) | 9.6 | 94.1 | 58.5 | $240.5 \pm 1.17$ | $63.1 \pm 3.0$ | $315.2 \pm 2.7$ | $322.8 \pm 3.4$ |
| Brewers spent grain (BS) | 2.6 | 51.2 | 487.7 | $215.2 \pm 4.27$ | $37.9 \pm 5.0$ | $47.1 \pm 2.3$ | $307.7 \pm 2.3$ |
| Corn silage (CS) | 7.3 | 81.1 | 189.0 | $230.0 \pm 7.09$ | $30.5 \pm 4.5$ | $101.7 \pm 1.0$ | $347.9 \pm 1.4$ |
| *Elodea nutallii* (EL) | 12.9 | 94.5 | 54.8 | $46.4 \pm 3.2$ | $21.2 \pm 6.7$ | $533.0 \pm 1.8$ | $344.7 \pm 2.5$ |
| Bran (BR) | 12.4 | 85.7 | 143.1 | $288.8 \pm 2.4$ | $23.3 \pm 3.9$ | $338.6 \pm 7.8$ | $206.2 \pm 4.3$ |
| Insect frass pilot plant (IF_PP) | 84.2 | 91.0 | 89.7 | $228.8 \pm 5.1$ | $33.9 \pm 5.2$ | $226.5 \pm 0.8$ | $421.1 \pm 5.2$ |

TS total solids; vs. volatile solids; FM fresh matter; * single sample.

## 2.2. Analytical Methods

Total solids (TS) and volatile solids (VS) were measured in accordance with DIN EN 12,880 (2001) [20] and DIN EN 12,879 [21]. The pH-value of digestate samples was measured with a pH device 3310 (WTW Wissenschaftlich-Technische Werkstätten GmbH, Weilheim, Germany). The Weender feed analysis of insect frass and ammonia nitrogen (NH4-N) and the total ammonia nitrogen (TAN) of the digestate were determined, as described in [22]. Once a week, fresh digestate samples were taken from CSTR digestion and centrifuged with 10,000 rpm for 10 min at 10 °C. Filtered samples (10 mL) of the supernatant liquid were used for the quantification of all volatile organic acids (VOA) and the ratio of VOA to total inorganic carbonate to calcium carbonate (VOA/TIC, gVOA/gCACO3) measurement in a Titration Excellence T90 titrator (Mettler-Toledo GmbH, Zurich, Switzerland).

## 2.3. Anaerobic Digestion Trials

### 2.3.1. Biochemical Methane Potential (BMP) Test

The six different insect frass samples were analyzed for specific methane yields (SMY) at lab-scale using the AMPTS2 BMP test system (Bioprocesscontrol, Lund, Sweden). BMP tests were carried out in accordance with the VDI guideline 4630 (2016) [23] under mesophilic conditions (39 ± 1 °C). The inoculum to substrate (ISR) ratio was approximately 3:1 (based on mass VS). Before the batch experiments, the AD reactors' headspace was flushed with nitrogen gas for about 2 min to assure anaerobic conditions. Each reactor contained approximately 2.5 gVS of insect frass and 400 g inoculum and was analyzed in triplicate. The SMY was standardized according to DIN 1343 [24] (dry gas, 273.15 K, 101.325 kPa). The BMP test ended after 38 days; the daily methane production had reduced to just 0.5% of the total biogas production for a minimum of 5 days. The pure inoculum was measured as a blank sample to determine the specific methane yield and to subtract this from the other samples. To monitor the inoculum performance, microcrystalline cellulose (MCC) was used as a reference substrate and the reference BMP confirmed a sufficient inoculum quality with 351 ± 11 mL/gVS. As inoculum served digestate (pH 7.8, VOA/TIC, Ammonia NH4-N 1.49 g/L) which was adapted to a wide range of substrate components, such as protein, fat, fiber, and carbohydrates, at a low OLR of 0.5 gVS/(L*d) over the duration of one year.

### 2.3.2. Semi-Continuous Anaerobic Digestion Tests

Two continuously stirred tank reactors (CSTRs; R1 and R2) in duplicate, each with a net volume of 15 L (10 L working volume) (Bräutigam Kunststofftechnik GmbH, Mohlsdorf-Teichwolframsdorf, Germany), were used for semi-continuous AD of insect frass derived from pilot plant operation (IF_PP). The main objective was assessing the AD process performance, stability, and methane production from IF_PP. The temperature was set at 39 °C using a thermostat (JULABO GmbH, Seelbach, Germany) and kept under mesophilic conditions (38 ± 1 °C) by recirculating hot water through the double-walled reactors. The reactors were continuously stirred (100 rpm) using a Stirrer 'RZR 2102 control' (Heidolph Instruments GmbH & Co.KG, Schwabach, Germany) located in the upper part of the reactors. The biogas volume was measured with a drum-type gas meter TG05/5 (Dr.-Ing. RITTER Apparatebau GmbH & Co. KG, Bochum, Germany), and the biogas quality was determined using a AwiFLEX (Awite Bioenergie GmbH, Langenbach, Germany). CSTR tests were conducted in accordance with the VDI guideline 4630 (2016) [23] as well. Methane and biogas yields were standardized, respectively (dry gas, 273.15 K, 1013.25 kPa). The fermentation experiments were accompanied by numerous analyses that were used for process characterization and monitoring, such as the dry matter organic dry matter analysis of the substrate and digestate samples, pH, ammonium, and volatile fatty acid concentration.

The general procedure for reactor operation at the DBFZ and detailed information on the accompanying analytics can be found in the literature reference [22]. For the CSTR experiment, the same inoculum was used as for the BMP tests (see Section 3.2). The

experiment was carried-out over 314 consecutive days with the same feeding frequency (once per day). After 5 days without feeding, reactors R1 and R2 were fed with DDGS (distillers' dried grains with solubles) pellets due to a delay in the supply with insect frass. On day 25, feeding of both reactors with IF_PP started. During start-up (Phase 1) the organic loading rate (OLR) was set to 1.0 g VS/L·d. Between days 52–140 (Phase 2), the OLR was increased to approximately 1.5 g VS/L·d and between day 141–173 (Phase 3) reduced to 0.7 g VS/L·d for 32 days due to process instability. Thereafter, the OLR was gradually increased from 0.7 to 1.5 and finally to 2.2 g VS/L·d until the end of the experiment between day 174–314 (Phase 4). When the final OLR of 2.2 g VS/L·d was reached, 30 g FM insect frass and 150 mL tap water were added daily to each digester. The hydraulic retention time (HRT) of ~80 days was kept constant over the first half of the experiment until day 130. Thereafter, the HRT was reduced to about 60 days until the end of the experiment. No additives, such as trace elements, were used. Detailed information about different feeding rates, OLR, and HRT are listed in Table 2.

**Table 2.** Overview of the reactor's setup during the AD experiment with IF_PP.

| Phase | Period (Day) | HRT (Days) | OLR (g VS/L·d) |
|---|---|---|---|
| Phase I | 0–52 | 80 | 1.0 |
| Phase II | 53–140 | 80 | 1.5 |
| Phase III | 141–173 | 60 | 0.7 |
| Phase IV | 174–314 | 60 | 1.5–2.2 |

Insect frass from pilot plant operation (IF_PP), hydraulic retention time (HRT), and organic loading rate (OLR).

### 2.4. Kinetic Evaluation

Two kinetic models were used to fit the experimental data of the BMP of the six different insect frass (i.e., EL, CS, BS, BR, ST, and IF_PP). These models were the first-order models and the modified Gompertz model, as given in Equations (1) and (2),

$$\beta(t) = \beta_0 \cdot \left[ 1 - e^{-kt} \right] \tag{1}$$

$$\beta(t) = \beta_0 \cdot e^{\left[ -e^{\left( \frac{\beta_m \cdot e}{\beta_0} \cdot (\lambda - t) + 1 \right)} \right]} \tag{2}$$

where $\beta(t)$ is the cumulative methane yield at time $t$ (mLCH4/gVS), $\beta_0$ is the maximum cumulative methane production predicted at a theoretically infinite digestion time (mLCH4/gVS), $k$ is the first order hydrolysis constant (1/days), $t$ is the time (days), $\beta_m$ is the maximum methane production rate (mLCH4/gVS·d), and $\lambda$ is the lag phase (days). In addition, model parameters and their uncertainties were estimated using the negative logarithm of the likelihood LL (Equation (3)) as the objective function with constant error variance [25],

$$LL = \frac{n \ln(2\pi\sigma^2)}{2} + \frac{\sum_{i=1}^{n} \left( \beta_i^{obs} - \beta_i^{est} \right)}{2\sigma^2} \tag{3}$$

where $n$ is the total number of experimental data, $i$ is an index, $\beta_i^{obs}$ represents the observed cumulative specific methane yield at time $t$, $\beta_i^{est}$ represents the estimated cumulative methane yield calculated with Equations (1) and (2), and $\sigma$ is the standard error. The model selection for the best fit to observed data was conducted using the Akaike information criterion (AIC) (Equation (4)).

$$AIC = -2 \ln(L_{max}) + 2P \tag{4}$$

where $L_{max}$ is the maximum likelihood and $P$ is the number of parameters included in the model. All parameters and their uncertainties were estimated using the subroutine "*optim*" from the statistical package R [26] using the L-BFGS-B algorithm.

### 2.5. Statistical Analysis

The data recorded after 32 days of AD obtained from the BMP test was analyzed using one-way analysis of variance (ANOVA) for comparing the specific methane yield (SMY) means among the six different insect frass samples (i.e., EL, CS, BS, BR, ST, and IF_PP). After one-way ANOVA, a post-hoc analysis with a Sidak post test for multiple comparison was performed. In addition, we used the Mann–Whitney rank sum test (Normality test, $p = 0.000$) for the data obtained from the semi-continuous experiment to compare the SMY values between reactors (R1 and R2). Descriptive statistics (mean and standard deviation) were computed for all the insect frass samples. All analysis was performed with Minitab V16.0 (Minitab Inc., State College, PA, USA). The statistical significance was set at $p \leq 0.05$.

## 3. Results and Discussion

### 3.1. Insect Frass Characteristics

As shown in Table 1, the total solid (TS) content of insect frass samples from lab scale rearing experiments was in a range between 2.6 and 12.9% FM while the TS of residue material obtained from the pilot plant operation was comparably dry with a TS of 84.2% FM. During lab scale rearing experiments, the humidity of the feed medium was controlled while in the pilot plant scale a certain drying of the feed medium is wanted. The lab scale separation of larvae and insect frass by wet sieving is easily possible while large-scale dry sieving is favored, as less handling effort is required, and a more transport-worthy insect frass is produced.

### 3.2. Effect of the Six Different Insect Frass Samples on Specific Methane Yield (SMY) from the BMP Test

The specific methane yield (SMY) of the tested insect frass samples ranged from 201 to 287 mL/g VS (Figure 1A) and is comparable to other residues from livestock farming, such as cattle manure (110–275 mL/g VS), pig manure (180–360 mL/g VS), and chicken manure (200–360 mL/g VS) [27]. A slightly lower SMY of approximately 177 mL/g VS was reported by Bulak et al. [28], measured in BMP tests with insect frass from Hermetia reared on residues from the fruit and vegetable industry in the form of carrot-beetroot marc.

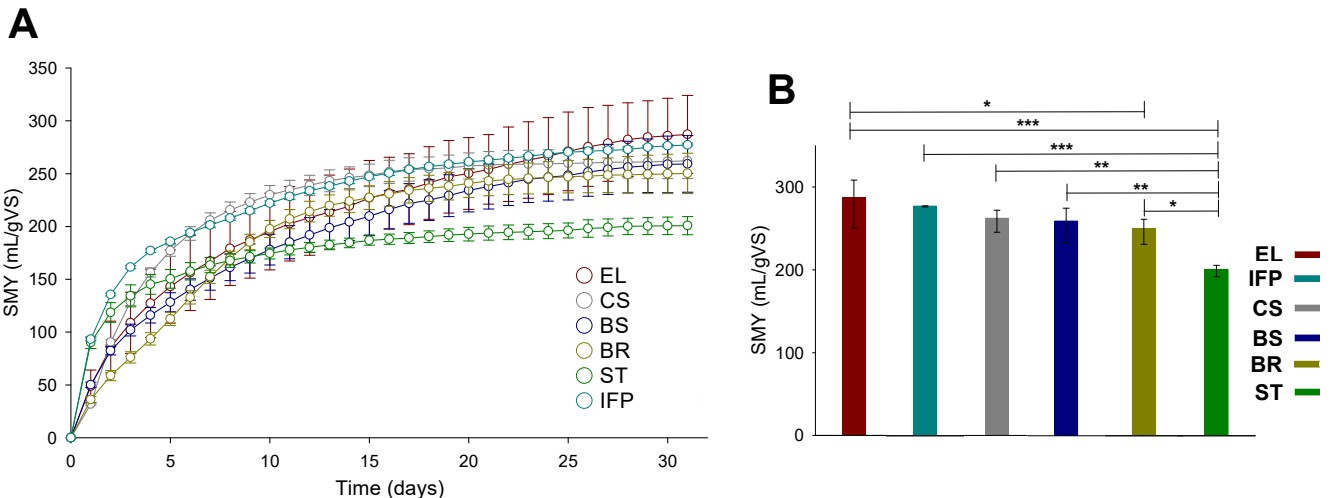

**Figure 1.** Specific methane yield (SMY) of the different insect frass samples. (**A**): *Elodea nutallii* (EL), corn silage (CS), brewers spent grain (BS), bran (BR), stillage (ST), and insect frass (IF_PP). Effect of different substrates on SMY (**B**). Data are mean ± standard deviation (*n* = 3) for each insect frass sample. * $p < 0.05$, ** $p < 0.01$. *** $p < 0.001$ (one-way ANOVA with Sidak post-test).

Our test results show the highest SMY for EL, followed by IF_PP, CS, BS, and BR, while the lowest SMY was shown by the insect frass sample ST with a value of 201 ± 8.6 mL/g VS. Results of the one-way ANOVA analysis indicated that at least one insect frass sample

was significantly different in the SMY mean values among the six different frass samples ($F$ = 5.833, $p$ = 0.006; Figure 1B). A post-hoc analysis for multiple comparison (Holm-Sidak method) showed that samples EL, CS, BS, BR, and IF_PP had significantly higher SMY versus the ST sample ($p$ < 0.05; Figure 1B), with increases of up to 30.07, 23.41, 22.53, 19.76, and 27.55%, respectively. These increases were probably due to differences in the characteristics of the insect frass samples used in the present study. For example, the IF_PP sample showed a lower fiber content than that of the ST (Table 1). Indeed, a negative effect on the SMY values would be expected from the increase in fiber fractions, in particular the ADF and ADL-like fraction [29].

The data also showed that only the EL sample had a significantly higher SMY compared to the BR sample, increasing SMY by about 12.85% ($p$ = 0.059, considered significant due to borderline significance). We also found that statistically similar SMY was observed among the EL, CS, BS, BR, and IF_PP insect frass samples (287 $\pm$ 36.8, 262 $\pm$ 16.9, 259 $\pm$ 26.9, 250 $\pm$ 19.1 and 277 $\pm$ 0.8, respectively; $p$ > 0.05) and were not significantly different from each other, except for the EL and BR samples (Figure 1B). This suggests that any change in SMY values may be attributed to differences in chemical composition among the six insect frass samples, likely due to the composition of the substrates used to previously feed the larvae. Overall, the SMY obtained from the BMP trials indicate, in the first instance, a good degradability of all insect frass samples used in the present study. After only about 30 days of the BMP test, gas formation was largely completed.

SMY from batch tests were similar for all insect frass samples and showed the potential suitability as AD feedstock. However, an economic evaluation would be necessary to assess the economic feasibility. There is, however, still very limited knowledge about methane production (i.e., SMY) from insect frass available that can support the findings of this study. Further research is required in order to validate the presented results and to extend the database of different insect frass materials.

### 3.3. Effects of the Different Insect Frass Samples on Estimated Model Parameters

Parameter estimates of the applied model structures (Equations (1) and (2)) are presented in Table 3. All models fit the observed data well with correlation coefficients ($R^2$) varying from 0.998–0.999; however, the first order model had the overall lowest AIC values in all the insect frass samples. Furthermore, parameter estimates of the lag phase duration ($\lambda$) in the modified Gompertz model were most often negative. Since the modified Gompertz model is only defined for positive parameter values ($\lambda \geq 0$), negative parameter estimates indicate that the model is not suitable for the description of measured methane production. Therefore, we selected the first order model as the best fit for the observed methane production of the insect frass samples: EL, CS, BS, ST, and IF_PP. The hydrolysis constant ($k$) obtained from the first-order kinetic model is mainly used to evaluate the substrate suitability and estimate the process rate-limiting stage. In this way, $k$ describes the velocities of degradation and methane production; therefore, a high $k$ represents high rates of degradation and methane production. In this study, our results provided evidence that high substrate biodegradation improved $k$, and thus improved the methane production rate and methane yield. The overall highest $k$ corresponded to the ST, IF_PP, and CS samples, while the lowest $k$ values were obtained for EL, BS, and BR. One possible explanation for the highest $k$ values of the samples is that there is a greater proportion of more easily degradable substances in these insect frass samples.

**Table 3.** Estimated parameters for the insect frass samples: *Elodea nutallii* (EL), corn silage (CS), brewers spent grain (BS), bran (BR), stillage (ST), and insect frass (IF_PP) for the first-order model and modified Gompertz model.

| Parameters | Insect Frass Samples | | | | | |
| --- | --- | --- | --- | --- | --- | --- |
| | EL | CS | BS | BR | ST | IF_PP |
| **Observed SMY** | $287 \pm 36.8$ | $262 \pm 16.9$ | $259 \pm 26.9$ | $250 \pm 19.1$ | $201 \pm 8.6$ | $277 \pm 0.8$ |
| **First order model** | | | | | | |
| $\beta_0$ (mL/gVS) | $280 \pm 4.5$ | $261 \pm 1.0$ | $257 \pm 3.8$ | $259 \pm 2.1$ | $190 \pm 2.2$ | $262 \pm 3.7$ |
| $k$ (1/day) | $0.13 \pm 0.01$ | $0.22 \pm 0.00$ | $0.13 \pm 0.01$ | $0.13 \pm 0.00$ | $0.38 \pm 0.03$ | $0.25 \pm 0.02$ |
| Correlation coefficient ($R^2$) | 0.9915 | 0.9985 | 0.9932 | 0.9983 | 0.9725 | 0.9761 |
| Akaike information criterion (AIC) | 247.10 | 183.49 | 236.95 | 191.03 | 243.25 | 266.63 |
| **Modified Gompertz model** | | | | | | |
| $\beta_0$ (mL/gVS) | $269 \pm 3.7$ | $256 \pm 1.2$ | $248 \pm 3.7$ | $248 \pm 1.3$ | $190 \pm 2.4$ | $259 \pm 2.7$ |
| $\lambda$ (days) | $-1.00 \pm 0.39$ | $-0.33 \pm 0.21$ | $-1.00 \pm 0.45$ | $-0.29 \pm 0.14$ | $-1.00 \pm 0.37$ | $-1.00 \pm 0.30$ |
| $\beta_m$ (mL/gVS·day) | $20.35 \pm 1.25$ | $34.13 \pm 1.93$ | $18.64 \pm 1.26$ | $21.65 \pm 0.52$ | $32.75 \pm 4.51$ | $33.25 \pm 2.73$ |
| Correlation coefficient ($R^2$) | 0.9787 | 0.9929 | 0.9825 | 0.9972 | 0.9563 | 0.9583 |
| Akaike information criterion (AIC) | 287.40 | 232.97 | 265.93 | 205.35 | 258.46 | 309.52 |

Specific methane yield (SMY), maximum cumulative methane production predicted ($\beta_0$), first order hydrolysis constant ($k$), maximum methane production rate ($\beta_m$), and lag phase ($\lambda$).

*3.4. Results of the Semi-Continuous Anaerobic Digestion Experiment*

The purpose of the fermentation test was to determine the maximum biogas potential and the process stability when using insect frass from a pilot plant operation (IF_PP) as a sole substrate. Figure 2 illustrates the AD performance of the two reactors R1 and R2, where the whole digestion period was divided into four phases: phase I (start-up, 0–52 day), II (53–140 day), III (141–173 day), and IV (174–314 day). It can be observed that R1 and R2 produced statistically similar SMY during phase I, II, and IV. Only during phase III did the digester show statistically significant differences according to the Mann–Whitney rank sum test ($p = 0.000$). During the course of the experiment, it was observed that the ammonium nitrogen ($NH_4$-N) concentration in both reactors increased continuously, which was likely due to the degradation of the protein-containing components of the insect frass (Figure 1A). According to Yenigün et al. (2013) [30], at a $NH_4$-N concentration of ~3 g/L, depending on the test temperature and pH value, an inhibition of the biogas process can already occur, which leads to an increase in fermentation acids, a decrease in pH, and thus a reduction in methane formation in the long-term test. This effect is very likely to be seen in Figure 1B at the end of phase II and during phase III. Only after a short-term reduction of the daily feeding amount (day 145 to 173) and an increase in the amount of water added to the substrate material (day 140), could the process be stabilized again.

In the second half of the experiment (phase IV), the organic loading rate (OLR) was increased in two steps from 0.7 to 1.4 and thereafter to 2.2 g VS/L·d, which led again to an increase in NH4-N concentration. However, the biogas process seemed to be already adapted to the substrate and higher NH4-N concentrations, as no further process instability was observed until the end of the experiment (phase IV).

According to Lalander et al. (2019) [31] and Abduh et al. (2022) [32], insect feeds with high protein content result in a higher insect biomass and protein yield. Therefore, common insect feed materials can contain comparatively high protein concentrations and, as a result, higher nitrogen levels may also be found in insect frass. Ammonium nitrogen (NH4-N) is a degradation product of organic nitrogen, such as proteins or urea during the AD process. In the digester liquid, $NH_4$-N is present as ammonium ions ($NH_4+$) and as free ammonia ($NH_3$). Increasing the pH or temperature results in a higher percentage of $NH_4$-N present as $NH_3$. It has been demonstrated that $NH_3$ is more toxic, as it can pass through the cell membrane, causing a proton imbalance and potassium deficiency [30]. According to Jiang

et al. [33], several research groups reported that the inhibitory $NH_4$-N concentration may range between 1.5 to 5.0 g/L. However, concentrations of up to 14 g/L are possible by the microbial adaptation of the digestion process. In the case of our study, an inhibition of the process biology can be expected during the end of phase II and beginning of phase III, which negatively affected the production of methane. Thus, the mixing of ammonia rich insect frass with other biogas substrates with a lower nitrogen concentration, such as whole plant silage could be an approach in order to stabilize the AD process in a practical application.

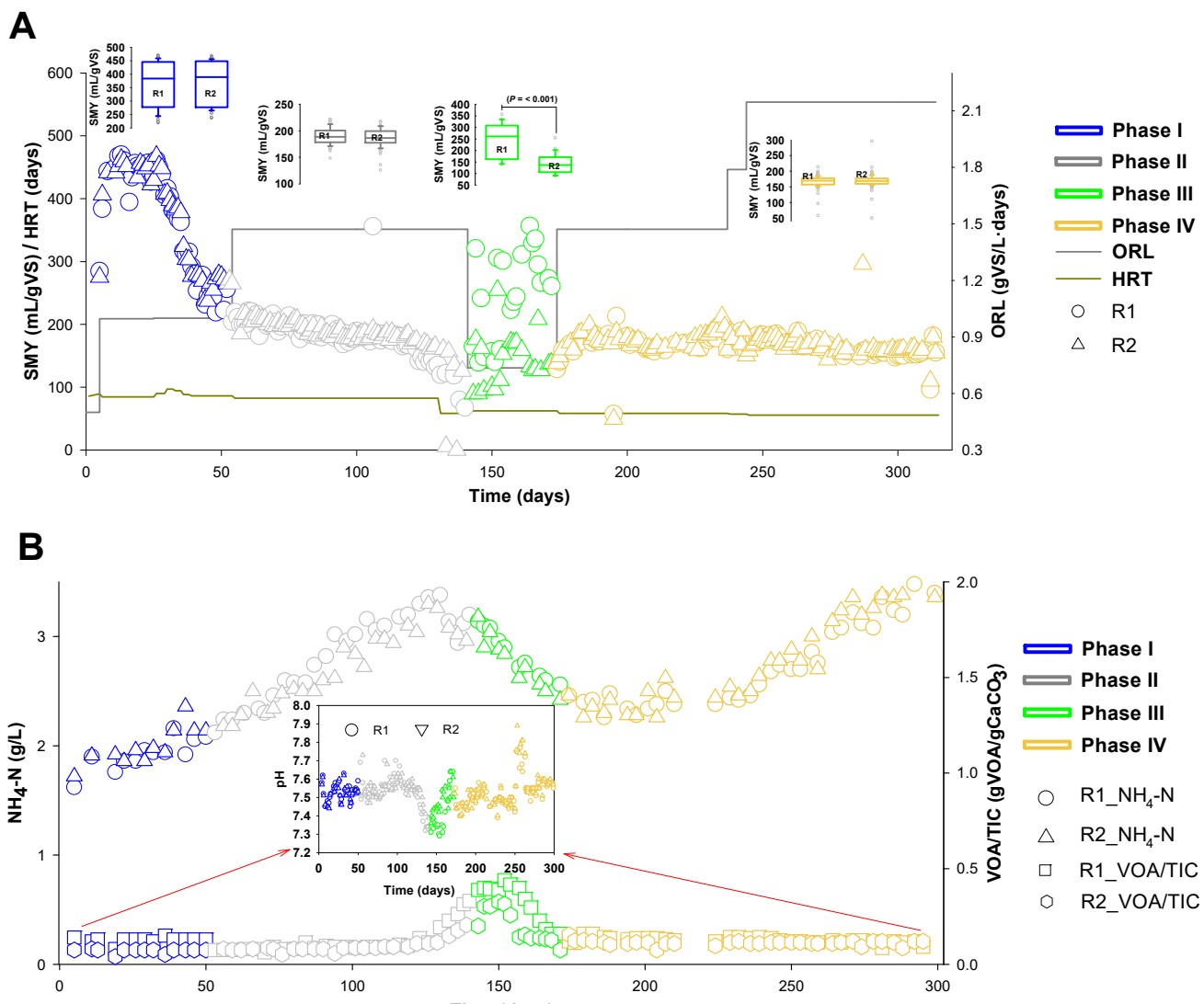

**Figure 2.** Anaerobic digestion (AD) process performance of the insect frass pilot plant (IF_PP) sample. Specific methane yield (SMY), organic loading rate (ORL), and hydraulic retention time (HRT) (**A**). Ammonium nitrogen ($NH_4$-N) concentration, ratio of volatile organic acids to total inorganic carbon (VOA/TIC) and pH (**B**).

The mean SMY measured in the semi continuous tests was in the range of $167 \pm 15$ mL/g VS and the mean methane content of the biogas was approximately 54%. Thus, the anaerobic digestion of insect frass in the CSTR trials also resulted in specific methane yields comparable to other agricultural residues from livestock farming (see Section 3.1). However, in comparison to the BMP tests the long-term digestion of insect frass from pilot plant operation (IF_PP) resulted in an approximately 38% lower SMY.

Weinrich et al. [34] pointed out that most studies comparing batch and continuous AD have reported lower SMY from continuous AD. A study from Ruffino et al., 2015 [35], determined a 24% lower SMY from the continuous AD of vegetable waste compared to the BMP results. Similarly, Zhang et al., 2013 [36], recorded 30% lower SMY from continuous AD of food waste compared to those of results from the BMP test. Holliger et al., 2017 [37], proposed that an extrapolation coefficient of 0.8 to 0.9 should be used to estimate the methane production of full-scale AD plants from BMP results of the substrates to be digested. According to [34] the continuous test systems will have a lower yield than the biogas potential and, in theory, also a lower yield than the BMP result at the same retention time. The causes stated are differences in the substrate degradation kinetics due to distinctions of the test systems and substrate material characteristics and potentially additional limitations. As an example, a sufficient supply of macro- and micronutrients can be assumed in the BMP test, while nutrient deficiencies or increasing concentrations of inhibiting substances in the digester medium can occur in long-term continuous digestion experiments.

Based on fresh matter, insect frass from pilot plant operations (IF_PP) achieved specific SMY of about 140 m$^3$/tFM (data not shown), which exceeds SMY of the common biogas substrate corn silage (CS) with 110 m$^3$/tFM [27]. Accordingly, one ton of the energy crop could be replaced by using one ton of insect frass from Hermetia rearing. Corn silage is the most frequently used biogas substrate of agricultural biogas plants in Germany, next to cattle slurry. Energy crop silage is mostly produced by agricultural biogas plant operators for the biogas plant demand, or bought in. In the last two drought years, 2021 and 2020, there was a shortage of corn silage supply in individual regions of Germany. The available reserves were mainly used for dairy farming and individual biogas plants and could not be fully utilized due to substrate shortages, resulting in a lower annual energy production and reduced profitability of the plants concerned. In the future, alternative agricultural residues, such as insect frass, could help to replace feedstock quantities of corn silage in existing biogas plants and reduce the demand for corn silage in years with a low feedstock supply.

The integrating of the insect farming process into the operation of existing biogas plants could lead to several synergetic effects. A process combination of insect farming and biogas production would enable comprehensive material and energetic biomass utilization. Further energetic synergy effects arise from the process combination through the use of waste heat from the biogas plant for heating the insect farm and for product drying. In Germany, more than 9000 biogas plants are in operation. These plants have an established raw material supply, the corresponding material handling, and provide large amounts of thermal energy on site. Insect production and the associated processing can be based on or aligned with this infrastructure. This could result in new value chains and business models for biogas plant operators which could help to increase insect protein production capacities.

As described in the introduction, sufficient sanitization of insect frass is required before the residual material can be used as soil amendment in agriculture. According to current knowledge, especially thermophilic digester systems with and without a downstream composting stage offer an effective hygienization method for waste biomass [38]. Particularly in the field of biowaste and sewage sludge treatment, the AD process is used to reduce the amount of waste and for waste hygienization. However, further research is required to evaluate the sanitization efficiency of the anaerobic digestion of insect frass. Additionally, the development of safe process chains that also include transport and residue handling at the biogas plant demands additional research.

## 4. Conclusions

Insect frass is a residue material of the insect rearing process and composes of feed remains and insect excrement. Anaerobic digestion can be an interesting waste treatment option with a potential for waste sanitization. By combining insect farming and biogas process electrical energy, biomethane as a biofuel can be produced from the waste material.

Further synergies arise from the utilization of exhaust heat of the biogas plant in the insect rearing process and product processing.

According to this initial study, insect frass represents a suitable biogas substrate with specific methane yields comparable to other residue materials from animal husbandry.

Insect frass from the pilot operation resulted in BMP tests in SMY of 277 ± 0.8. mL/g VS. In comparison to the BMP tests, long-term anaerobic digestion resulted in approximately 38% lower methane yields in a range of 167 ± 15 mL/gVS with a mean methane content of the biogas of approximately 54%. During the digestion trial, an increase in the ammonium concentration was observed, which can lead to process instabilities. Therefore, the anaerobic digestion of insect frass with co-substrates which have a lower nitrogen content can be recommended.

**Author Contributions:** The manuscript was written through the contributions of all authors. Conceptualization and methodology, H.W.; validation, D.G.I.; statistical and kinetic analysis, D.G.I.; visualization, D.G.I.; formal analysis, D.R.J.; writing—original draft preparation, H.W.; writing—review and editing, D.G.I. and D.R.J.; supervision, H.W.; project administration, H.W.; funding acquisition, H.W. All authors have read and agreed to the published version of the manuscript.

**Funding:** This work was supported by funds of the Federal Ministry of Education and Research under the innovation support program (Neue Produkte für die Bioökonomie) (FKZ: 031B0338A/B, CIP; www.dbfz.de/cip) and KMU Innovativ Biotechnologie (FKZ: 031B1111A/B, BioLube; www.dbfz.de/biolube).

**Data Availability Statement:** Not applicable.

**Acknowledgments:** The author wants to acknowledge the technical staff of the DBFZ laboratory and especially Haase, Susan Hoffmann, Yue Wang, and Chen Fu for their contributions to the experimental and analytical work.

**Conflicts of Interest:** The authors declare no conflict of interest. The funders had no role in the design of the study; in the collection, analyses, or interpretation of data; in the writing of the manuscript; or in the decision to publish the results.

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
