# Peer review of "Biogas Production from Residues of Industrial Insect Protein Production from Black Soldier Fly Larvae Hermetia illucens (L.): An Evaluation of Different Insect Frass Samples"

_processes, doi:10.3390/pr11020362_

Round 1

Reviewer 1 Report

Comments

The manuscript reports on the biogas production from residues of industrial insect protein and evaluates the effects of different insect frass samples. It demonstates the insect frass utilization as feedstock for anaerobic digestion in both lab-scale fermentation trials and semi continuous anaerobic digestion trials. In addition,  the data of long term semi-continuous AD trials in CSTR digesters helped to evaluate the feedstock digestibility and process stability which may be informational to the readers. The manuscript is well-written and the experimental details are clearly described. All in all, the manuscript is interesting and worth publication while the authors should address the following comments in the revision.

1.     The degree of novelty of this manuscript, as well as the advance added to the respective area, must be clearly stated.

2.     Please simplify the introduction of this manuscript. For instance, only highlight the most relevnet background and focus the on the importance of this work.

3. Please stated the research development of insect frass in biogas production.

4. Please elaborate on the advantage and disadvantage of biogas production from insect frass, including economic and commercial aspects.

5.    The results of the study should be compared to other recent studies.

6. The title of the manuscript needs to be concise with carefully consideration of words like “black soldier fly larvae (BSF) Hermetia illucens (L.) ”.

7.   Page 7 Line 314-316 needs Ref or other supports for the explanation. (Also, check the whole manuscript carefully for typos like “explanaition”)

Author Response

Thank you very much for your comments which helped to improve the manuscript. I have answered your comments in the attached pdf file. Kind regards Harald Wedwitschka

Reviewer 2 Report

This manuscript evaluates BSFL frass for biomethane production through batch and semi-continuous AD tests. Overall, the study provides some useful insights the BSFL frass for mass reduction and bioenergy production. However, the logic of the results and discussion is not clear. Therefore, the manuscript needs Major Revision before final acceptance. The weaknesses points include:

·       The abstract is weak and does not show the paper novelty and objectives.

 ·       The introduction is sketchy. The state-of-the-art is not properly described in the Introduction section. The Introduction can be improved through circular bioeconomy approaches. The following recently published relevant articles on applied insect farming for waste reduction and bioenergy production   should also be included and discussed in a number of studies, e.g., https://doi.org/10.1016/j.envres.2022.113708 and https://doi.org/10.1016/j.renene.2022.02.029

 ·       In Section M&M., are great need of careful revision and organization. You should provide some photos and all information about the frass derived from pilot plant operation (IFP), BMP and semi-continuous AD tests.

·       The characteristics data in the table1 must be provided with the standard deviation value

·       In section results and discussion, the manuscript needs more discussion in order to completely investigate the frass derived from pilot plant operation (IFP) mainly composed of cereal grain for insect production, which makes their use expensive because it is mainly used as poultry feed. How can this be dealt with economically?

 ·       The highest BMP yield recorded with frass from Elodea genus (EL), however the lowest yield with stillage (ST), What is the main reason behind these results?

·       English should be further improved.

Author Response

Thank you very much for your comments which helped to improve the manuscript. I have answered your comments in the attached pdf file. 

Kind regards

Harald Weditschka

Round 2

Reviewer 2 Report

I thank the authors for the good response to the suggestions. I recommend accepting the manuscript in its current form.

Author Response

Thank you very much for your helpful comments in the first review round. The suggestions have been implemented in the manuscript. English language was improved by native speakers.
